# EyeBench: Predictive Modeling from Eye Movements in Reading

**Omer Shubi**[V]* **David R. Reich**[U][P]* **Keren Gruteke Klein**[V] **Yuval Angel**[V]
**Paul Prasse**[P] **Lena Jäger**[U] **Yevgeni Berzak**[V]

[V] Technion – Israel Institute of Technology, [U] University of Zurich, [P] University of Potsdam
{shubi,gkeren,yuval.angel}@campus.technion.ac.il
{davidrobert.reich,lenaann.jaeger}@uzh.ch
paul.prasse@uni-potsdam.de
berzak@technion.ac.il
* Equal contribution

## Abstract

We present EyeBench, the first benchmark designed to evaluate machine learning models that decode cognitive and linguistic information from eye movements during reading. EyeBench offers an accessible entry point to the challenging and underexplored domain of modeling eye tracking data paired with text, aiming to foster innovation at the intersection of multimodal AI and cognitive science. The benchmark provides a standardized evaluation framework for predictive models, covering a diverse set of datasets and tasks, ranging from assessment of reading comprehension to detection of developmental dyslexia. Progress on the EyeBench challenge will pave the way for both practical real-world applications, such as adaptive user interfaces and personalized education, and scientific advances in understanding human language processing. The benchmark is released as an open-source software package which includes data downloading and harmonization scripts, baselines and state-of-the-art models, as well as evaluation code, publicly available at https://github.com/EyeBench/eyebench.

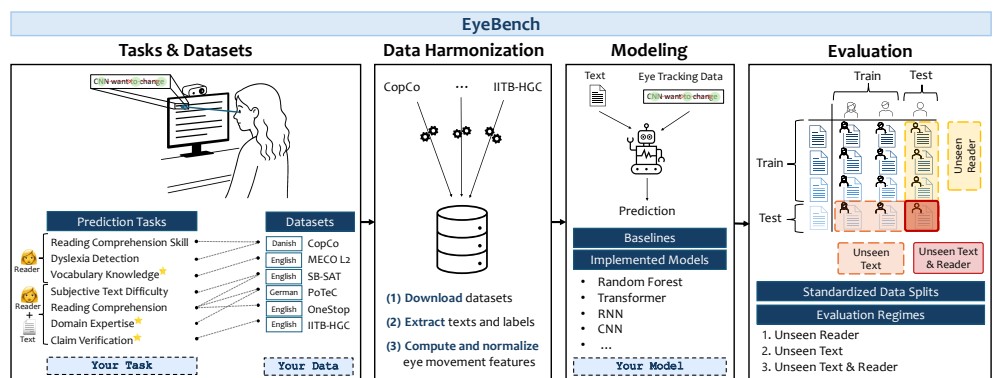

Figure 1: Overview of EyeBench v1.0. The benchmark curates multiple datasets for predicting reader properties (🧑), and reader-text interactions (🧑+📄) from eye movements. ⭐ marks prediction tasks newly introduced in EyeBench. The data is preprocessed and standardized into aligned text and gaze sequences, which are then used as input to models trained to predict task-specific targets. The models are systematically evaluated at three different levels of generalization to new readers, texts, or both. The benchmark supports the evaluation and addition of new models, datasets and tasks.

39th Conference on Neural Information Processing Systems (NeurIPS 2025) Track on Datasets and Benchmarks.

# 1 Introduction

Understanding how humans process language is a long-standing scientific question and key to advancing machine intelligence. One of the most powerful sources of insight into human language processing comes from eye tracking during reading. Eye movements provide a fine-grained, temporally detailed record of how readers interact with text, reflecting cognitive processes such as lexical access, syntactic parsing, semantic integration, and discourse comprehension [1, 2, 3]. Yet, despite their rich informational content and widespread use in psycholinguistic research, these data remain strikingly underexploited in machine learning research and applications.

In this work, we introduce EyeBench, the first benchmark that provides an infrastructure for systematically tracking progress on the challenges of modeling eye movements in conjunction with text, aiming to promote machine learning research in this emerging area. Eye tracking while reading data represents a unique form of spatio-temporal multimodal data, combining a static sequence of words (the stimulus text) with a dynamic time series of eye movements located directly over this text, posing a rich and distinctive challenge for machine learning. The relationship between the two modalities is structured and complex: it is driven by linguistic (e.g., lexical frequency, syntactic complexity, discourse structure, writing system [4, 5, 6, 7]) and non-linguistic properties of the text (e.g., layout [8] and font [9, 10]), individual differences among readers (such as linguistic proficiency, reading proficiency and cognitive strategies [11, 12, 13]), and by the nature of the reading task itself (ordinary reading for comprehension, skimming, proofreading, information seeking and others [14, 15, 16, 17, 18]).

Furthermore, unlike many other domains studied in machine learning, eye tracking datasets are non-i.i.d. in a structured way: each data point is tied to both a reader and a textual item. Typically, each reader reads several textual items, and each textual item is presented to multiple readers. Consequently, model generalization needs to be evaluated in different regimes, each relevant to different types of potential applications: unseen readers, unseen textual items, or both. Existing approaches in multimodal machine learning only partially address these modeling and evaluation complexities, and so far, relatively few models have been proposed for this domain, indicating a clear need and opportunity for modeling innovation.

Predictive modeling using eye movements in reading holds significant potential for both impactful real-world applications and contributions to science. Real-time predictions from gaze behavior can enhance interactive systems by enabling adaptive interfaces that respond to user attention and cognitive state. Such predictions are especially pertinent for education, where gaze-informed systems could personalize reading instruction, detect comprehension difficulties, and support second language learners. Harnessing machine learning and NLP towards these goals can substantially expand the scope and utility of technologies developed in these areas of research and provide fertile ground for modeling innovation. At the same time, predictive models from eye tracking data offer a powerful tool for scientific discovery. Accurate machine learning models of gaze behavior can help disentangle the underlying cognitive processes that govern online language comprehension. By revealing which aspects of the linguistic input and the reader most strongly shape eye movement patterns during reading, machine learning models may help refine cognitive theories of language processing.

Historically, progress in the domain of eye tracking for reading has been limited by the scarcity of large, high-quality, and diverse eye tracking datasets, as well as by a lack of standardized evaluation protocols. However, the field is now at a turning point: recent efforts have produced datasets of sufficient quality, quantity, and task diversity to support meaningful predictive modeling. With this in mind, the benchmark introduced here aims to provide an accessible, standardized entry point into the domain of eye tracking and reading for the machine learning and NLP communities.

The tasks defined in EyeBench are highly challenging. Despite recent progress, none of the existing models are close to a level of predictive performance that would make them viable for real-world applications. This underscores the need for innovative approaches that better capture the structured, multimodal nature of eye tracking data during reading. Importantly, EyeBench is designed to support both the development of task-specific models, as well as general-purpose models that integrate eye movement behavior and linguistic input more broadly. The benchmark introduced here provides a streamlined framework for evaluating both types of models in a standardized and comparable way, facilitating progress in developing models that are both effective and generalizable. It further supports the addition of new tasks and datasets on which models can be tested.

## 2 Background

### 2.1 Eye Movements in Reading and Cognitive Processes

Reading behavior is characterized by a distinct pattern of eye movements of alternating *fixations* and *saccades*. Fixations are periods of time, lasting 200–250 ms on average, during which the eyes remain relatively stationary and visual information is acquired and processed [1, 19, 20]. In contrast, saccades are rapid, ballistic movements lasting approximately 20–80 ms that shift the gaze from one fixation to another [21]. During saccades, visual perception is largely suppressed, a phenomenon known as saccadic suppression [20]. While most saccades during reading are *forward saccades* that move the gaze forward along the text, several types of deviations occur. *Regressions* involve backward movements to earlier parts of the text [1]. *Skips* occur when words are bypassed without a direct fixation, whereas *refixations* involve additional fixations within the same word [3]. Figure 2 presents a schematic example of eye movements during reading, illustrating the sequence of fixations and saccades for a single sentence.

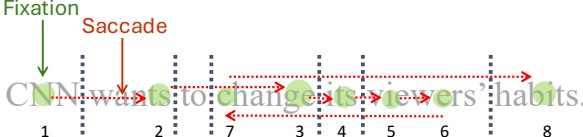

Figure 2: Illustration of eye movements during reading. Green circles represent fixations, where visual information is gathered, and red arrows represent saccades, which are shifts in gaze between fixation points. Dotted vertical lines denote word boundaries.

Such sequences of fixations and saccades offer a powerful lens into the cognitive processes that unfold during reading. Extensive research over several decades has established strong links between gaze behavior, such as fixation durations, saccade patterns, and regressions, and the underlying mechanisms of attention, linguistic processing, and comprehension, as well as the moment-by-moment cognitive demands imposed by the text [22, 23, 24, 25].

With the rise of modern machine learning and NLP, studies have leveraged eye tracking data to predict a variety of cognitive states and linguistic attributes of readers. These include linguistic background [26, 27, 28], language proficiency [29], subjective text readability [27], and reading comprehension performance [27, 30, 31, 32]. Eye movements have also been used to decode higher-level reading goals [33, 34, 35] and repeated reading interactions [36]. Together, these advances have paved the way for computational approaches that aim to infer a wide range of real-time cognitive states and linguistic skills directly from patterns of eye movements during reading.

### 2.2 Eye Tracking Data Representation

Modern video-based eye tracking systems capture eye movements with high temporal precision, often at millisecond-level resolution, and high spatial precision, allowing researchers to distinguish gaze locations at the level of individual words or even letters. The resulting raw gaze data comprises time-stamped sequences of gaze positions, which are typically processed to identify and segment gaze events: fixations and saccades. The resulting sequence of alternating fixations and saccades is often referred to as the eye movement *scanpath*. The fixations and saccades that comprise the scanpath can be used to extract measures such as the duration of a single specific fixation. They can further be aggregated to derive *word-level* reading measures, such as the average fixation duration or number of fixations on a specific word. Word-level measures as well as other metrics describing the scanpath (e.g., overall ratio of progressive vs regressive saccades, variance in fixation durations) can in turn be further aggregated for larger textual units such as a passage, or for specific interest periods of the experiment, such as an experimental *trial*.

These different representations, ranging from the raw signal to global statistics of eye movement measures, enable the study of reading behavior across multiple granularities of eye movement information. The models implemented in EyeBench, presented in Section 3.5, cover a range of such representational approaches from prior literature. EyeBench will enable researchers to systematically explore the effectiveness of existing approaches and investigate new representations of eye movements in reading across different tasks.

## 3 EyeBench

EyeBench is a benchmark for multimodal predictive modeling using eye movements in reading, covering a wide range of discriminative tasks of both theoretical importance to cognitive science and practical relevance for user-facing applications. It provides the machine learning community with a challenge and an opportunity for modeling innovation with a unique, and thus far underexplored, type of multimodal data comprising eye movements and text. EyeBench considers eye tracking-while-reading problems which take as input the eye movement recordings of a reader, and optionally the stimulus text, and predict either a property of the reader (e.g., whether the reader is affected by developmental dyslexia), or their interaction with the text (e.g., the reader's comprehension of a given stimulus text). It consists of an end-to-end infrastructure that covers data preparation and model evaluation for this kind of inference tasks.

The EyeBench pipeline, presented in Figure 1, includes seven prediction tasks over six publicly available datasets. A data loader downloads and harmonizes the benchmark datasets into a shared format. The benchmark includes implementations of baselines and 12 state-of-the-art models for each task-dataset pair. It further includes standardized training, validation, and test sets for each dataset and task, covering three different model generalization regimes, accompanied by evaluation scripts and results. This pipeline forms an accessible entry point to the domain of eye movements in reading, which allows machine learning researchers to focus on modeling, rather than on data preparation. It further supports continuous and meaningful community progress on the benchmark tasks through standardized evaluation. In addition to providing a testbed for new models, EyeBench supports community contributions of new datasets and tasks to the benchmark. EyeBench is released publicly as an open-source software package in the official benchmark repository, along with extensive documentation. Usage notes for different use cases of the benchmark are provided in Appendix A.

### 3.1 Problem Setting

The prediction tasks in EyeBench take as input data from a single experimental *trial*, during which a single participant $R$ reads a single textual passage $T = \langle w_1, w_2, \ldots, w_n \rangle$ composed of a sequence of words. As the participant reads the passage, their eye movements are recorded, resulting in a sequence $E_T^R = \langle e_1, e_2, \ldots, e_m \rangle$ of gaze events (fixations and/or saccades). Each gaze event $e_i$ includes spatial and temporal information (e.g., position, duration) and can be linked to a word in the text. The input may further consist of an auxiliary trial-specific text presented to the participant $T_{task} = \langle w_1, w_2, \ldots, w_m \rangle$, such as a reading comprehension question, with or without corresponding eye movements $E_{T_{task}}^R$.

Each task in EyeBench is formulated as a predictive modeling problem: given the input tuple $\left(T, E_T^R, T_{task}, E_{T_{task}}^R\right)$ from a single trial, the goal is to predict a target variable $y^R$ or $y_T^R$, where $y^R$ is a property of the reader (🧑), and $y_T^R$ is a characteristic of the reader-text interaction (🧑+📄). This corresponds to modeling the conditional distribution $p\left(y^R \mid T, E_T^R, T_{task}, E_{T_{task}}^R\right)$ or $p\left(y_T^R \mid T, E_T^R, T_{task}, E_{T_{task}}^R\right)$, respectively.

Note that prior work on the prediction of reader properties (🧑) has often aggregated data from multiple trials in the task input [26, 29, 37, 38]. Here, we choose the more flexible, and typically more challenging, single-trial setting to enforce consistency and comparability with reader-text interaction tasks. In reader-text interactions (🧑+📄), the prediction $y_T^R$ is always for a single behavioral response (e.g., an answer to a single reading comprehension question) rather than for an aggregation measure of several such responses [27, 39]. Finally, we note that eye movements can also be used for prediction tasks about the text itself (📄). Such tasks are currently not included in EyeBench.

### 3.2 Tasks

EyeBench includes a sample of seven tasks based on their *theoretical importance* to the fields of cognitive science and the psychology of reading and their *practical relevance* to real-world applications. An additional key selection criterion is the availability of passage-level *publicly available datasets* for the task (see further details in Section 3.3). Appendix D.2 lists tasks from the literature that are currently not included in EyeBench. We note that several of these tasks, such as *reading goal* [35] and *repeated reading* prediction [36], meet the EyeBench inclusion criteria, and are expected to be added in future versions of the benchmark.

The seven tasks in EyeBench include a standardized, single-trial formulation of established tasks that were addressed in prior work, as well as newly introduced tasks which we study here for the first time. As mentioned above, the tasks can be broadly divided into two categories, 🙇: inferring language-related properties or skills of the reader $y^R$, and 🙇+📄: inferring properties of a reader's interaction or engagement with a given text $y_T^R$. Each of these categories includes both classification and regression tasks. In all cases, the task input contains eye movements $E_T^R$ and the stimulus passage $T$ over which the eye movements were collected.

### 3.2.1 Reader Properties 🙇

**Reading Comprehension Skill** Assessing reading comprehension skill is a core component of educational practice. However, traditional assessments, which rely on reading comprehension questions, are costly and time-consuming to develop, often ad hoc, and introduce task-related overhead. In contrast, eye movement–based assessments could potentially offer a cheaper, more implicit, and less intrusive alternative testing methodology. Crucially, eye movements are a process measure which, in contrast to traditional product measures, more directly reflects language comprehension processes as they unfold over time, free from strategic meta-level reasoning about question answering. From a cognitive science perspective, this task enables researchers to investigate how differences in reading comprehension ability manifest themselves in gaze behavior. Prediction of reading comprehension skill is an existing task that has been recently explored using linear regression models [31], CNNs and RNNs [27, 39], as well as DenseNets and random forests on feature embeddings learned with contrastively pretrained GANs [40].

Reading Comprehension Skill is a regression task supported by CopCo [41, 42, 43]. The prediction target is the reader's general reading comprehension competence, assessed independently through a standardized test administered outside of the eye tracking task, and represented as a continuous variable ranging from 1 to 10.

**Vocabulary Knowledge** Assessment of linguistic proficiency is vital for monitoring second language (L2) acquisition progress in education, as well as for the scientific study of individual differences in language processing. Prior work has demonstrated that language proficiency is reflected in eye movements in reading [13, 44, 45], and can be predicted from the eye tracking record [29]. Here, we focus on the estimation of vocabulary knowledge, a key component of linguistic knowledge, which correlates well with general linguistic proficiency [46].

The Vocabulary Knowledge task is newly introduced in EyeBench and is supported by the MECO L2 dataset [47, 48]. It is a regression task aimed at predicting the score obtained in the LexTALE English vocabulary test [46], which ranges from 0 to 100.

**Dyslexia Detection** Approximately 3–7% of the children in a given cohort are affected by developmental dyslexia [49, 50], a genetically [51, 52, 53] and neurologically [54, 55] based reading disorder characterized by a specific and significant impairment in the acquisition of reading skills [56] which persists in adulthood. A substantial body of research in cognitive psychology has shown that readers with dyslexia exhibit distinct eye movement patterns compared to readers without dyslexia [57, 58, 59]. In machine learning, a growing body of work has been using eye tracking data to automatically distinguish between dyslexic and non-dyslexic readers using traditional classifiers [37, 60, 61, 62], as well as neural networks [38, 63, 64, 65]. This task is theoretically relevant for understanding how reading difficulties associated with dyslexia are reflected in gaze behavior, and practically valuable for enabling scalable, non-intrusive screening methods that could complement traditional diagnostic procedures.

The Dyslexia Detection classification task is implemented in EyeBench using the CopCo dataset [41, 42, 43]. The binary target label is based on a clinically confirmed diagnosis of dyslexia.

### 3.2.2 Reader-Text Interactions 🙇 +📄

**Reading Comprehension** Assessing how well a person has understood a specific text is crucial in many educational and professional real-world contexts. In contrast to the Reading Comprehension Skill task, which targets *general* reading comprehension proficiency, the Reading Comprehension task focuses on *text-specific* understanding: the goal is to predict whether a reader will correctly answer a given comprehension question about a specific text, based on their eye movement patterns during

reading. The relationship between eye movements and reading comprehension has long been a central topic in the psychology of reading and psycholinguistics, with numerous studies demonstrating that eye movements reflect online comprehension processes [22, 66, 67, among others]. From this cognitive perspective, the task enables studying effective versus ineffective reading strategies and how they manifest themselves in gaze behavior. Practically, the prediction of a reader's comprehension of a given text can have multiple applications, including the real-time monitoring of comprehension in educational tools, adaptive content delivery based on user understanding, or support for content accessibility. Previous work has addressed a number of variants of this task using a range of modeling approaches, including linear regression models [31], Fisher kernels derived from generative models of eye movements [30], RNNs and CNNs [27, 39], feature embeddings learned with GANs [40], and transformers [32]. Overall, these studies suggest that the task is highly challenging and emphasize the need for more advanced and effective modeling techniques.

In EyeBench, the Reading Comprehension task is implemented as a classification task using three datasets, OneStop [68], SB-SAT [39], and PoTeC [69]. In addition to eye movements over a passage $E_T^R$ and the passage $T$, the model input includes $T_{task}$: a reading comprehension question that was presented to the participant after reading the passage. The prediction output is a binary label indicating whether the participant answered the question correctly. We note that the specific task formulation was not studied with PoTeC [70] and SB-SAT [71, 39, 27, 64], where prior studies only categorized participants based on aggregated scores across questions, binarized into high and low scores relative to the median across participants.

**Subjective Text Difficulty**   Readers differ in how difficult they perceive a given text to be. This perception can depend on various factors, including language and reading skills, motivation, familiarity with the topic, as well as factors such as alertness or fatigue. Understanding how readers experience the complexity of a text is essential for enhancing digital accessibility and building adaptive learning systems that tailor content to individual needs. While psychological research has shown that objectively measured text difficulty correlates with distinct reading patterns, little is known about how subjectively perceived difficulty shapes eye movement behavior. Inferring subjectively perceived text difficulty could thus offer new opportunities for psychological reading research, while also enhancing practical applications for automatic text selection, personalization, and adaptive content presentation based on perceived difficulty. Previous approaches to this task include CNNs and RNNs [27, 39], as well as feature representations learned via contrastively pre-trained GANs [40].

The Subjective Text Difficulty task is supported in EyeBench by SB-SAT [39]. It is operationalized as a regression task. The target is a post-reading subjective difficulty rating of the text on a Likert scale.

**Domain Expertise**   Understanding a reader's level of expertise in the domain of a text can be crucial for various educational and professional settings. It offers the opportunity to deepen our understanding of how prior knowledge interacts with cognitive processes involved in reading, and help disentangle the relative influence of different factors shaping gaze behavior. From a practical standpoint, this task enables personalized content selection by tailoring materials to a reader's background knowledge, without requiring explicit assessments. Recent research has shown that domain knowledge systematically influences eye movement behavior during reading [72], and demonstrated the feasibility of a related task, predicting whether a reader's academic background matches the domain of the text read based on gaze data [70].

Inferring a reader's domain expertise from eye movements is a new task first introduced in this benchmark. The task is supported by the PoTeC dataset [69]. Domain Expertise is a classification task. The target is a binary label (high vs low domain expertise), based on whether after reading the passage, the reader correctly answered three background knowledge questions that are closely related to, but do not overlap with, the content of the stimulus text.

**Claim Verification**   Claim verification is a specific variant of a reading comprehension task that assesses a reader's ability to critically evaluate whether a given claim is supported by a text. It taps into higher-order comprehension skills, such as reflective reading and critical engagement, and also indicates the degree to which the reader is attentively processing the content. Recent research has demonstrated that, in question answering and claim verification tasks, humans tend to selectively attend to task-relevant parts of the text [18, 73]. Predictive modeling of this process can reveal what eye movement patterns are associated with successful information extraction and critical reading.

From a practical perspective, the ability to infer claim verification from eye movements can support educational tools that promote critical thinking and reading strategies.

Claim Verification is a new task introduced in EyeBench. It is implemented using the IITB-HGC dataset [73]. Claim Verification is a classification task. The input consists of a short passage $T$ and a preceding claim $T_{task}$ (either supported or unsupported by the passage), along with the reader's eye movements $E_T^R$ and $E_{T_{task}}^R$ on both $T$ and $T_{task}$. The prediction target is a binary label indicating whether, after having read the passage, the participant correctly judged the claim as supported by the passage or not.

### 3.3 Datasets

EyeBench includes six passage-level public datasets that support the tasks described above. An additional key selection criterion is the availability of scanpath information (the complete sequence of fixations and optionally saccades) in the dataset, or raw data from which this information can be computed. The complete dataset inclusion criteria are provided in Section B.1 of the Appendix. Based on these criteria, the included datasets in EyeBench are: the Ordinary Reading portion of OneStop [68], SB-SAT [39], PoTeC [69] MECO L2 [47, 48], CopCo [41, 42, 43] and IITB-HGC [73]. All the datasets consist of recordings of eye movements using an SR Eyelink or Portable Duo eyetracker at a sampling rate of 1000Hz or higher. Note that a single dataset can support multiple tasks, and a given task may be supported by multiple datasets.

Table 1 provides key dataset information, including the language of the texts, the participants' linguistic background, supported tasks, and summary statistics. Further dataset descriptions are provided in Appendix B.2. Table D in the Appendix presents 41 additional datasets that are not included in the benchmark, alongside the specific inclusion criteria that they do not meet. We stress that while these datasets are currently not included in EyeBench, they are highly valuable resources, which can be used for various purposes, including pretraining models evaluated on EyeBench and other tasks, and exploring research questions beyond the scope of the benchmark.

Table 1: Datasets used in EyeBench. "Participant Group" indicates the linguistic status of the readers in the language of the text: L1 (native speakers) or L2 (non-native speakers), and whether they are affected by developmental dyslexia. "Number of Words" is the number of words in the underlying textual corpus. "Number of Task Instances" is the total number of input-output samples for the task.

| Dataset | Text Language | Participant Group | Number of Participants | Number of Words | Total Number of Fixations | Tasks | Number of Task Instances |
|---|---|---|---|---|---|---|---|
| OneStop Ordinary Reading [68] | English | L1 | 180 | 19,428 | 1,114,034 | Reading Comprehension | 9,718 |
| SB-SAT [39] | English | L1, L2 | 95 | 2,622 | 263,032 | Reading Comprehension Subjective Text Difficulty | 1,900 380 |
| PoTeC [69] | German | L1 | 75 | 1,895 | 403,775 | Reading Comprehension Domain Expertise | 2,700 900 |
| MECO L2 [47, 48] | English | L2 | 1,098 | 1,646 | 2,409,160 | Vocabulary Knowledge | 9,493 |
| CopCo [41, 42, 43] | Danish | L1, L2, L1 with dyslexia | 57 | 32,140 | 397,883 | Reading Comprehension Skill Dyslexia | 4,782 4,782 |
| IITB-HGC [73] | English | L1, L2 | 5 | 53,528 | 163,910 | Claim Verification | 2,500 |

### 3.4 Data Loading and Harmonization

One of the key bottlenecks for research with eye movements is the different formats in which eye tracking datasets are released, and the lack of standardized preprocessing pipelines for these data. EyeBench addresses this challenge by introducing a standardized pipeline that automatically downloads and preprocesses datasets into a single unified format. The pipeline is designed in a modular way, allowing for easy integration of various preprocessing steps. It currently includes a unified process of computing eye-movement measures, linguistic features of the text, and alignment of tokenized text with fixation- and word-level eye movement measures. After applying the preprocessing pipeline, all datasets are transformed into the same standardized format. Appendix B.2 provides a detailed description of the preprocessing steps, including dataset-specific modifications.

## 3.5 Implemented Models

We provide a comprehensive set of 12 implemented models from prior literature, nine of which are neural models, and three are traditional machine learning models. This set covers most of the currently existing models that combine eye movements and text for predictive modeling of the reader or their interaction with text. We currently do not include architectures designed for the prediction of eye movements in reading, such as Eyettention [74] and SP-EyeGAN [71].

The neural models include fixation-level and word-level architectures that integrate eye movements with or without textual input. The models are **AhnRNN** and **AhnCNN** [39], **BEyeLSTM** [27], **PLM-AS** [75], **PLM-AS-RM** [38], **RoBERTEye-W** and **RoBERTEye-F** [32], **MAG-Eye** [32], and **PostFusion-Eye** [32]. The traditional machine learning models are **Logistic Regression** [31], **SVM** [76] and **Random Forest** [77]. These models use trial-aggregated gaze features. While all the models have been introduced in prior work, they are adjusted and applied to new tasks in EyeBench. Further details on the models, including model backbones, are presented in Appendix C.1.

To meaningfully assess the performance of predictive models from eye movements, it is crucial to compare them to informative baselines that do not use eye movement information. In EyeBench, we include six such baselines. For classification tasks we include a **Random** baseline and **Majority Class** predictor. For regression tasks we use the target variable's **Mean** and **Median**. Two additional baselines apply to both types of tasks. The first is **Reading Speed**, which can be computed without eye tracking from the total reading time spent on the text, and is often correlated with the target label of the prediction task. The second represents a class of baselines which consists of **Text-Only** models that can capture task-informative statistics for textual items and participants. Here, we implement such a baseline using **RoBERTa** [78], which encodes the text without the eye movements. Table 2 summarizes the feature types used by the implemented models and baselines.

Table 2: Feature types used by each model. Eye movement features are divided into three levels of granularity: **Saccades/Fixations** (e.g., the duration of a specific fixation), **Words** (e.g., the average fixation duration on a given word), and **Trial** (e.g., average fixation duration across all the words in the trial). Text features are divided into: **Linguistic** word properties (e.g., word frequency), contextual word **Embeddings**, and information about the **Layout** of the text (e.g., screen position coordinates or line number of a given word).

| Model | Eye movement features | | | Text features | | |
| --- | --- | --- | --- | --- | --- | --- |
| | Saccade/ Fixation Level | Word Level | Trial Level | Linguistic | Embeddings | Layout |
| Majority Class / Chance | - | - | - | - | - | - |
| Reading Speed | - | - | ✓ | - | - | - |
| Text-Only RoBERTa | - | - | - | - | ✓ | - |
| Logistic Regression [31] | - | - | ✓ | - | - | - |
| SVM [76] | - | - | ✓ | - | - | - |
| Random Forest [77] | ✓ | - | ✓ | ✓ | - | - |
| AhnRNN [39] | ✓ | - | - | - | - | - |
| AhnCNN [39] | ✓ | - | - | - | - | - |
| BEyeLSTM [27] | ✓ | ✓ | ✓ | ✓ | - | - |
| PLM-AS [75] | ✓ | - | - | - | ✓ | - |
| PLM-AS-RM [38] | ✓ | ✓ | - | - | ✓ | - |
| RoBERTEye-W [32] | - | ✓ | ✓ | ✓ | ✓ | ✓ |
| RoBERTEye-F [32] | ✓ | ✓ | ✓ | ✓ | ✓ | ✓ |
| MAG-Eye [32] | - | ✓ | ✓ | ✓ | ✓ | ✓ |
| PostFusion-Eye [32] | ✓ | ✓ | ✓ | ✓ | ✓ | ✓ |

**Model Training and Hyperparameter Tuning** For each dataset, we use a $k$-fold cross-validation procedure, splitting the data into training, validation, and test sets, where the number of splits is determined per dataset. For OneStop, we use $k = 10$ splits following previous work on reading comprehension prediction [32]. For the remaining datasets, we use $k = 4$ splits due to their smaller number of participants or texts. Stratification across labels is used for classification tasks.

We perform hyperparameter tuning separately for each data split. Models are optimized to minimize the validation set loss using mean squared error for regression tasks and cross-entropy for classification

tasks. Further details on the training protocol and specific hyperparameter search spaces used for each model are provided in Appendix C.2.

## 3.6 Evaluation

**Evaluation Regimes**   As mentioned above, differently from many other domains in machine learning, the samples in eye-tracking-while-reading datasets are not independently and identically distributed (i.i.d.) in a structured way: each participant reads multiple passages, and multiple participants read the same passage. It is therefore essential to account for these dependencies during evaluation. To this end, each test set is divided into three subsets to evaluate different aspects of model generalization:

- **Unseen Reader**: The text passage appeared in the training data, but the reader did not.
- **Unseen Text**: The reader appeared in the training data, but the text passage did not.
- **Unseen Reader & Text**: Neither the reader nor the passage appeared in the training data.

The *Unseen Reader* setup is relevant to scenarios such as classical assessments where prior data exists for the materials but not for the test-taker. The *Unseen Text* setting mirrors real-world use cases such as individualized digital learning platforms or gaze-based assistive technologies that adapt to new content for a known user. The *Unseen Reader & Text* regime represents the most demanding and flexible scenario, which enables the broadest range of potential applications, including systems that are agnostic to both user and content. We further note the possibility of implementing an evaluation where both the test reader and the test text passage appear in the training set, but not paired. We do not include it in EyeBench, as it provides limited insight into the models' ability to generalize.

**Task-specific performance metrics**   We report *AUROC* and *balanced accuracy* for classification tasks, and *RMSE*, *MAE*, and $R^2$ for regression tasks. Task-level model performance for each task-dataset pair is computed over the aggregation of all the test sets.

**Global performance metrics**   To further facilitate model comparisons, we compute an overall *Average Normalized Score* for each model, obtained by macro-averaging task-level scores for the model across all task-dataset pairs and metrics, ensuring that each task contributes equally regardless of the dataset size [79]. Furthermore, we compute the overall *Mean Rank* for each model, determined by averaging the task-level ranks across all task-dataset pairs [80, 81].

# 4   Model Benchmarking Results

Model evaluations for version `v1.0` of EyeBench are released on the official benchmark repository. These include (i) summary views of global performance metrics (ii) task-specific performance metrics (iii) results for each task and dataset by evaluation regime (*unseen reader*, *unseen text*, *unseen reader & text*). Future evaluations will be with respect to specific benchmark releases, where future versions of the benchmark are expected to include new models, tasks, and datasets.

In the initial `v1.0` release of the benchmark, there is no conclusive evidence that any single model consistently outperforms the others across tasks, datasets or evaluation regimes. In several settings, some of the baseline models perform competitively with the current state-of-the-art approaches. At present, none of the tasks reach a level of performance that would be relevant for real-world applications, leaving ample room for future progress on the benchmark tasks.

# 5   Discussion

We introduce EyeBench to encourage and facilitate progress in the development of machine learning models that decode rich cognitive signals from eye tracking data during reading. Differently from many other machine learning benchmarks which are reaching saturation, the performance of current models on the EyeBench tasks leaves much room for improvement. We envision that such improvements will be driven by modeling innovation, which can in turn inform other domains that involve multimodal modeling.

To support continued progress in this space, additional data collection efforts are needed. To guide such efforts, we emphasize the importance of open, well-documented, large-scale, and high-quality data, spanning diverse languages, populations, text genres, and reading interactions. We advocate for the release of raw data and fixation-level gaze event data alongside any aggregated measures at the word and trial levels. These finer-grained signals are crucial for enabling temporal modeling and for investigating the sequence dynamics of reading behavior.

Our current focus is on high quality data collected with state-of-the-art eye tracking equipment in-lab. With the rapid development and improvement of low cost eye tracking technology, it will likely be possible to collect larger scale eye tracking for reading data, in more naturalistic settings using lower cost devices such as glasses, webcams, as well as augmented reality (AR) and virtual reality (VR) devices. We aim for future iterations of EyeBench to include such data, which will in turn facilitate deployment in domains like educational technology, adaptive reading tools, and cognitive monitoring platforms. In sum, EyeBench provides a foundation for bridging cognitive science and machine learning through eye tracking data. We envision that it will catalyze innovation not only in model design and data collection but also in practical applications that bring language technologies closer to human-like understanding and interaction.

**Ethical and Societal Considerations**   Data collection, analysis, and predictive modeling with eye movement data involve multiple ethical and societal considerations. We first note that all the datasets in the benchmark were collected under institutional IRBs, and follow strict data anonymization protocols. The prediction tasks of EyeBench fall under the intended uses of the datasets. Importantly, this benchmark focuses on decoding *language-related* cognitive information, with the goal of enabling applications that offer positive societal impact, such as language assessment, educational tools for language learning, and improved digital accessibility. We note, however, that eye movements in reading can be attempted to be used for reader identification and for prediction of extra-linguistic reader characteristics such as gender or age, which we discourage. We further emphasize that future deployment of user-facing applications that involve eye movements in reading requires explicit consent from the user for their eye movements to be collected and analyzed for the specific purposes of the applications. With that in mind, we envision that EyeBench will catalyze community building around fundamental research and applications with eye movements in reading that will benefit society.

## Acknowledgments and Disclosure of Funding

This work was partially funded by COST Action MultiplEYE (Action number: CA21131), supported by COST (European Cooperation in Science and Technology), the Swiss National Science Foundation under grant IZCOZ0_220330 (EyeNLG, PI: Lena Jäger), and the Israel Science Foundation under grant 1499/22 (PI: Yevgeni Berzak).

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

# A Usage Notes

EyeBench is a maintained and versioned framework for benchmarking predictive models that process eye movements in reading. It was designed with three primary use cases in mind. First, researchers interested in tasks that predict either reader-specific attributes or properties of the reader's interaction with the text can use the benchmark to identify modeling strategies best suited to their goals. Second, it enables systematic benchmarking of new models across seven well-defined and meaningful tasks on six harmonized datasets, using evaluation protocols that reflect realistic deployment scenarios. Third, the benchmark provides access to 12 state-of-the-art models and six strong baselines, and supports benchmarking of new datasets that broaden the diversity of reading scenarios and participant populations.

Each release (e.g., EyeBench-vX.X) represents a fully reproducible snapshot of:

1. Implemented and fine-tuned models.
2. Harmonized datasets with standardized data splits across evaluation regimes.
3. Evaluation protocols with individual and aggregated metrics, including statistical testing.

We encourage contributions of new datasets, new prediction tasks, and new models to future releases of EyeBench, following the contribution protocols listed below. All benchmark code, metadata, and submission templates are publicly available at `github.com/eyebench/eyebench`. Users are encouraged to consult the latest version of the EyeBench documentation to verify any updates to the entry points, dependencies, or usage instructions.

## A.1 Benchmark your model

The model benchmarking protocol is as follows:

1. Model logic should be placed in `src/models/YOUR_MODEL_NAME.py`, following the structure of existing classes (e.g., inheritance from `BaseDLModel`/`BaseMLModel` for neural and machine learning models accordingly). Specifically, neural models should implement a `forward` function and a `shared_step` method that, given a batch of data, returns a loss, logits, and labels of that batch. Scikit-learn-based [82] machine learning models can optionally override the default `fit` and `predict` methods.
2. Define default parameters in `src/configs/models/<dl/ml>/YOUR_MODEL_NAME.py`.
3. Hyperparameter search spaces should be defined in `src/run/multi_run/search_spaces.py`. All hyperparameter tuning in Eye-Bench is performed using *triple cross-validation*. Existing search spaces are defined in `src/run/multi_run/search_spaces.py`. For further best practices in model tuning, we recommend consulting the Deep Learning Tuning Playbook [83].
4. Verify model compatibility with benchmark datasets using `src/run/multi_run/model_checker.sh`.

## A.2 Contribute your model

To contribute your model to EyeBench, it should meet the following criteria:

- It must be compatible with the datasets provided by EyeBench.
- The used hyperparameter grid must be provided.
- An associated publication that details the model architecture and training procedure must be provided.

## A.3 Benchmark your dataset

EyeBench can be used as an exploratory tool to accelerate research on novel tasks that use eye-tracking-while-reading data as input. By providing easy access to a suite of models, researchers can quickly deploy and test these models on new datasets or in new experimental contexts.

This allows to:

1. **Identify promising modeling strategies:** By observing which existing SOTA or baseline models perform well (or poorly) on a newly introduced problem or dataset, researchers can quickly hypothesize which classes of modeling strategies (e.g., sequence vs trial-level feature-based models) are most likely to succeed.

2. **Establish initial performance:** Using the reference models helps to rapidly establish an initial performance benchmark for a new problem, providing a starting point before investing significant resources in developing a completely novel architecture.

3. **Analyze model behavior:** The framework includes tools for analyzing the internal behavior of the reference models, allowing researchers to explore how different features (e.g., text, previous fixations, reader characteristics) are weighted by the model. This can help to generate new psychological or cognitive hypotheses about the reading process itself.

To add a new dataset:

1. Eye movement data and optionally textual data should be manually stored in `data/YOUR_DATASETNAME/` or integrated into pymovements.

2. Specify any preprocessing steps in `src/data/preprocessing/preprocess_data.py` and `src/data/preprocessing/dataset_preprocessing/YOUR_DATASETNAME.py`. Note, this requires a single file containing all "trials", you can achieve this using `src/data/preprocessing/union_raw_files.py`.

3. Define loading logic in `src/data/datasets/YOUR_DATASET.py`. See example code snippet 1.

4. Add a corresponding `src/data/datamodules/YOUR_DATASET_datamodule.py`.

5. Add a corresponding configuration in `src/configs/data.py` that specifies relevant hyper-parameters, such as the stratification variable or the maximum number of fixations in a single trial. Further define a task class `YOUR_DATASET_TASK(YOUR_DATASET)` per task that the dataset supports, in which you specify the target variable (e.g., the prediction column) and any task-specific preprocessing. See example code snippet 2.

Listing 1: Adding a new dataset

```
1  # src/data/datasets/YOUR_DATASET.py
2  from src.data.datasets.base import ETDataset
3  from src.configs.constants import DatasetNames
4
5  class YourDataset(ETDataset):
6      """
7      Dataset class for YOUR_DATASET.
8
9      Args:
10         args: Configuration or arguments for dataset loading.
11         stage (str): Dataset stage.
12             One of: 'train', 'val', 'test'
13         regime_name (str): Evaluation regime.
14             One of: 'unseen_text', 'unseen_reader',
15                     'unseen_reader_unseen_text'
16     """
17
18     def __init__(self, args, stage: str, regime_name: str):
19         super().__init__(args, stage)
20
21     def __getitem__(self, idx: int):
22         item = self.data.iloc[idx]
23
24         # Prepare text representation
25         text_inputs = self._prepare_text_inputs(item)
26
27         # Extract multi-level eye-tracking features
28         word_features = self._extract_word_features(item)
29         fixation_features = self._extract_fixation_features(item)
30         trial_features = self._extract_trial_features(item)
```

```
31
32          return {
33              'text_inputs': text_inputs,
34              'word_level_features': word_level_features,
35              'fixation_level_features': fixation_level_features,
36              'trial_level_features': trial_level_features,
37              'label': item['label']
38          }
```

Listing 2: Defining and registering a data–task pair.

```
1  # src/configs/data.py
2
3  from dataclasses import dataclass, field
4  from src.configs.data_args import DataArgs
5  from src.configs.utils import register_data
6  from src.constants import DatasetLanguage, PredMode
7
8  @register_data
9  @dataclass
10 class YourDataset(DataArgs):
11     """Base configuration for YOUR_DATASET."""
12
13     # --- Dataset metadata ---
14     dataset_name: str = 'YOUR_DATASET'
15     text_language: str = DatasetLanguage.ENGLISH # or GERMAN, DANISH,
16         etc.
17     # --- Stratification for train/val/test splits ---
18     stratify: str = 'target_variable_name'
19
20     # --- Data processing parameters ---
21     max_scanpath_length: int = 500
22
23 @register_data
24 @dataclass
25 class YourDataset_TRC(YourDataset):
26     """Task-specific configuration for reading comprehension."""
27
28     task: PredMode = PredMode.TRC
29     target_column: str = 'is_correct'
30     class_names: list[str] = field(
31         default_factory=lambda: ['Incorrect', 'Correct']
32     )
33     max_q_len: int = 30  # Maximum question length in tokens
34     max_tokens_in_word: int = 12  # For subword tokenization
```

### A.4   Contribute your dataset

To contribute your dataset as one of the benchmark datasets, your dataset must meet the following criteria:

1. Ensure that the data can be downloaded automatically via pymovements [84].

2. Integrate functionality within src/data/preprocessing/get_data.sh.

3. Maintain compatibility with models supplied by EyeBench.

4. Ensure compliance with the evaluation protocols.

5. Integrate performance metrics into aggregate score, introduced in Section 3.6.

## B EyeBench Datasets

### B.1 Selection Criteria

We use the following criteria for including datasets in the benchmark:

- Supports an EyeBench task.
- Passage level texts (not single sentences).
- Collected with a high quality eye tracker: sampling rate of at least 500 Hz, and typical calibration error below 0.5°.
- Publicly available scanpath data in the form of gaze events (fixations and, optionally, saccades), or raw eyetracking records from which gaze events can be computed.[1]
- Publicly available stimulus texts.
- An alignment between the scanpath data and the textual stimuli.

### B.2 Dataset Descriptions and Preprocessing Steps

**OneStop** The OneStop Eye Movements corpus [68] is a large-scale eye-tracking-while-reading dataset where native English (L1) participants read English newswire articles. Each article is divided into 4-7 text passages with a total of 162 passages. Each passage has three possible reading comprehension questions. The dataset includes four reading regimes: ordinary reading for comprehension, information-seeking reading, repeated reading, and information-seeking in repeated reading. The ordinary versus information-seeking manipulation is between participants, while repeated reading is within-participant. In EyeBench, we use the *ordinary reading* portion of the corpus, which includes 180 participants, each reading 54 passages, and answering a single comprehension question after each passage. Prior work used this dataset for *reading goal* decoding [34, 35], and prediction of *reading comprehension* [32] and *repeated reading* [36].

We used the fixation and word-level reading measures provided by the authors of the dataset, and computed the additional features required by the models in EyeBench from the provided data.

**SB-SAT** The **S**tony **B**rook **S**cholastic **A**ssessment **T**est (SAT) [39] contains eye-tracking-while-reading data from 95 participants who each read four passages taken from the reading comprehension section of the SAT exam. After reading each passage, participants' text comprehension was assessed using five original SAT questions. The dataset also includes participant-level subjective difficulty ratings for each text passage, and participant-level general reading comprehension scores, approximated by averaging the scores across all four passages. Previous work with SB-SAT addressed prediction of *general reading comprehension*, *text difficulty*, and *native speaker identification* [39, 27, 71].

Using optical character recognition, we extracted the stimulus texts from the original image files provided by the authors. Each extracted passage was then reviewed and manually corrected. Following text extraction, we computed all linguistic and eye movement features used in EyeBench. When a specific feature could not be computed due to missing or incompatible data, its value was approximated (e.g., saccade length was estimated from the distance between adjacent fixations), or if approximation was not possible, imputed as zero. In contrast to prior work, which treated eye-movement recordings from separate pages of a text as independent observations, we concatenated the text and corresponding eye-tracking data across all pages of a text passage to form a single instance per passage. For the reading comprehension task, each text-question pair represents one instance. We therefore copied the aggregated eye-movement sequence for each text passage and appended the corresponding comprehension question to the associated stimulus text.

**PoTeC** The **Po**tsdam **Te**xtbook **C**orpus (PoTeC) [69] consists of 75 German native speakers reading 12 scientific texts from physics and biology textbooks. A subset of the participants are experts in one of the two disciplines. After each text, participants answer three text comprehension questions

---

[1]Note that datasets that provide only trial- or word-level aggregated reading measures (e.g., SR Data Viewer "Interest Area Report"), without providing the full sequence of fixations and saccades (e.g. SR Data Viewer "Fixation Report") are not included, as they do not support fine-grained temporospatial modeling central to the benchmark tasks.

and three background knowledge questions, targeting text-based and general domain knowledge, respectively. PoTeC was previously used to predict whether a reader's academic discipline matches the domain of the text read based on gaze data [70].

We used the fixation and reading measures provided by the authors. For the reading comprehension task, each text-question pair represents one instance. Thus, we copied the eye-movement sequences for each text and appended the corresponding comprehension question to the associated stimulus text.

**MECO L2**    The L2 portion of the **M**ultilingual **E**ye-tracking **CO**rpus is a large-scale eye-tracking dataset from English L2 readers with 20 different native language backgrounds. We use the combination of Wave One [47] and Wave Two [48] of the dataset. The corpus includes eye movement recordings from 542 L2 English participants in Wave One and 660 participants in Wave Two. Each participant read 12 texts. The dataset was previously used for *native language identification* from eye movements [28].

We utilized the word-level reading measures and fixation data provided with the dataset. Moreover, we aligned fixations with the text to compute additional word-level features. Saccade length was approximated as the distance between consecutive fixations. Features that could not be computed or reliably approximated were imputed with a value of zero.

**CopCo**    The **Cop**enhagen **Co**rpus of Eye Tracking Recordings from Natural Reading of Danish Texts [41, 42, 43] comprises data from 58 native Danish-speaking participants reading texts in Danish. Participants are divided into three groups: Neurotypical L1 readers, L1 readers affected by developmental dyslexia, and neurotypical L2 readers. The dataset includes both scores obtained from text comprehension questions administered during the reading experiment and a separate reading comprehension assessment conducted independently of the eye-tracking session, used as a measure for general comprehension. Prior work used this dataset for predicting *text comprehension*, *general reading comprehension*, and *dyslexia*.

We utilized the SR Research Data Viewer interest area and fixation reports provided by the authors. To compute the additional features used in EyeBench, we aligned these reports with the stimulus text. Extractable features were computed, and missing features were approximated where possible. For example, saccade length was estimated as the distance between consecutive fixations. When a feature could not be computed or reasonably approximated, its value was imputed as zero.

**IITB-HGC**    The **I**ndian **I**nstitute of **T**echnology **B**ombay – **H**allucination **G**aze **C**orpus [73] comprises eye-tracking data from five annotators reading 500 claim–context pairs from the FactCC dataset [85]. Readers rated whether the content was hallucinated and whether they agreed with the initial annotation. Unlike the other datasets that report fixation locations in on-screen x- and y-pixel coordinates, IITB-HGC only provides a mapping between fixation locations and the words they were on. Thus, we use word indices as an approximation of the fixation position. When a feature could not be computed or reasonably estimated, its value was set to zero.

# C Implemented Models

## C.1 Models

**Neural models**

- **AhnCNN** [39]: A CNN model that relies solely on eye-movement data, represented by fixation sequences (x and y screen coordinates), fixation durations, and pupil size.
- **AhnRNN** [39]: An RNN-based variant of AhnCNN using the same input features.
- **BEyeLSTM** [27]: A deep learning model combining sequential fixation data and global gaze statistics via an LSTM followed by a linear projection layer.
- **PLM-AS** [75]: An RNN processing word embeddings that have been reordered to reflect the temporal order of fixations.
- **PLM-AS-RM** [38]: An RNN combining fixation-ordered word embeddings with eye-tracking reading measures..
- **RoBERTEye-W** [32]: A transformer that integrates word embeddings and word-level eye-tracking features at the input layer.
- **RoBERTEye-F** [32]: A fixation level variant of RoBERTye-W.
- **MAG-Eye** [32]: A transformer architecture adding word-level gaze features into intermediate transformer layers using a Multimodal Adaptation Gate mechanism.
- **PostFusion-Eye** [32]: A model combining RoBERTa-based word representations and CNN-extracted fixation features via cross-attention into a shared latent space.

**Text Backbones**: Models leveraging pretrained language models use the RoBERTa architecture [78], specifically the HuggingFace implementations of `roberta-large` for English datasets and `FacebookAI/xlm-roberta-large` [86] for non-English datasets. MAG-Eye, RoBERTEye-W, RoBERTEye-F, and PostFusion-Eye modify the transformer by integrating gaze information into its input, intermediate, or output layers, whereas PLM-AS and PLM-AS-RM use fixed RoBERTa embeddings as inputs to recurrent networks. In contrast, AhnCNN, AhnRNN, and BEyeLSTM do not rely on a language model.

**Traditional machine learning models**

- **Logistic Regression**: A logistic regression variant of the linear regression model of [31], with identical features.
- **SVM**: An SVM using global eye movement features from [76].
- **Random Forest**: A random forest using features based on [77]. [77] rely on features derived directly from raw gaze data, whereas the datasets used in our benchmark are based on segmented eye movement events. To adapt the approach from [77] to this setting, we compute a range of statistical aggregation functions (e.g., min, max, mean, skewness, etc.) over all available numerical features extracted from the benchmark's event-based data. These features include both those related to fixated interest areas and those computed from individual saccades and fixations.

For regression tasks, we use the corresponding regression variants of these models, namely **Linear Regression**, **Support Vector Regression (SVR)**, and **Random Forest Regressor**, with identical features.

## C.2 Training, Hyperparameters and Feature Normalization

All neural networks were implemented in PyTorch Lightning [87]. The machine learning models are implemented in scikit-learn [82]. Following [27], the AhnRNN, AhnCNN and BEyeLSTM models were trained for up to 1000 epochs with early stopping after 50 epochs. Following [32], the remaining neural models were trained for up to 10 epochs with early stopping after 3 epochs, and a linear 10 % warm-up schedule for the learning rate. To address class imbalance in the classification tasks, we use inverse class frequency weighting during training. Training was conducted on GPUs with 40-48 GB memory.

**Feature normalization**   Each eye-movement feature was standardized to zero mean and unit variance using statistics computed on the training set, consistent with prior work [32].

**Hyperparameter search space**   We performed hyperparameter tuning using a model-specific grid search, based on the search space proposed by the models' authors. The hyperparameter search space for each model is determined through a two-stage procedure. For existing tasks and models from the literature, when the original authors provide a recommended hyperparameter grid, we based the grid search on it. If the authors report best-found hyperparameters, we include those values within the search space. If the search space is limited or underspecified, we expand it. The following specifies the search space used for each model:

- **AhnRNN:** Learning rate $\in \{1e-5, 3e-5, 1e-4, 1e-3\}$, dropout rate $\in \{0.1, 0.3, 0.5\}$, hidden dimension $\in \{25, 50\}$, and number of LSTM layers $\in \{1, 2, 4\}$.

- **AhnCNN:** Learning rate $\in \{1e-5, 3e-5, 1e-4, 1e-3\}$, dropout rate $\in \{0.1, 0.3, 0.5\}$, and hidden dimension $\in \{40, 80\}$.

- **BEyeLSTM:** Learning rate $\in \{1e-3, 3e-3, 1e-2\}$, dropout rate $\in \{0.1, 0.3, 0.5\}$, hidden dimension $\in \{64, 128\}$, and embedding dimension $\in \{4, 8\}$ .

- **PLM-AS:** Learning rate $\in \{1e-5, 3e-5, 1e-4, 2e-4\}$, number of LSTM layers $\in \{1, 2\}$, LSTM dropout rate in $\{0.1, 0.3, 0.5\}$, and layer freeze $\in \{\texttt{True, False}\}$.

- **PLM-AS-RM:** Learning rate $\in \{1e-5, 3e-5, 1e-4, 2e-4\}$, LSTM hidden size $\in \{10, 40, 70\}$, and layer freeze $\in \{\texttt{True, False}\}$.

- **Text-Only RoBERTa, RoBERTEye-W, RoBERTEye-F and PostFusion-Eye:** Learning rate $\in \{1e-5, 3e-5, 1e-4\}$, eye projection dropout rate $\in \{0.1, 0.3, 0.5\}$, and layer freeze $\in \{\texttt{True, False}\}$.

- **MAG-Eye:** Learning rate $\in \{1e-5, 3e-5, 1e-4\}$, dropout rate $\in \{0.1, 0.3, 0.5\}$, layer freeze $\in \{\texttt{True, False}\}$, and MAG injection index of gaze information into the transformer $\in \{0, 23\}$.

- **Logistic Regression (Classifier):** Regularization strength $C \in \{0.1, 1.0, 5.0, 10.0, 50.0, 100.0\}$, and penalty $\in \{\texttt{l2, None}\}$.

- **Linear Regression (Regressor):** Intercept fitting $\in \{\texttt{True, False}\}$.

- **Support Vector Machine (Classifier):** Kernel $\in \{\texttt{rbf, linear}\}$, regularization $C \in \{0.1, 1.0, 10.0, 100.0\}$, and gamma $\in \{\texttt{scale, auto}, 0.1, 0.01, 0.001, 0.0001\}$.

- **Support Vector Regressor:** Kernel $\in \{\texttt{rbf, linear}\}$, regularization $C \in \{0.1, 1.0, 10.0, 100.0\}$, and gamma $\in \{\texttt{scale, auto}, 0.1, 0.01, 0.001, 0.0001\}$.

- **Random Forest (Classifier/Regressor):** Number of estimators $\in \{10, 100, 1000\}$, max depth $\in \{3, 6, 9\}$, min samples per split $\in \{2, 4, 8\}$, min samples per leaf $\in \{1, 0.01, 0.02\}$, and max features $\in \{\texttt{sqrt, log2, None}\}$.

# D   Datasets and Tasks not Included in EyeBench

## D.1   Datasets

Below are 41 existing eye-tracking-while-reading datasets which currently do not meet all the benchmark selection criteria. We list them below by their primary exclusion criterion.

**Texts not available** (datasets where the stimuli were not released):

- *Swedish Dyslexia* [37]
- *InDiCo* [88]
- *Binge Reading* [89]
- *GazeBaseVR* [90]
- *Alzheimer* [91]
- *Not Batting an Eye* [92]
- *ETSA I* [93], *ETSA II* [94]
- *IITB 1* [95], *IITB 2* [96], *IITB 3* [97]
- *DEMONIC* [98]
- *DMORPH* [99]
- *MQA-RC* [100]
- *MECO MO* [101]
- *Provo* [102]
- *WebQAmGaze* [103]

**Only word-level measures** (fixation-level information not released):

- *RastrOS* [104]
- *TECO* [105]
- *GECO* [106]
- *Dundee* [107]
- *Hahn & Keller* [108]
- *HKC Sentence* [109]
- *HKC Paragraph* [109]
- *Potsdam Allahabad Hindi Eyetracking Corpus* [110]
- *TURead* [111]

**Only sentences** (single-sentence stimuli rather than passages):

- *CELER* [112]
- *ZuCo* [113]
- *ZuCo 2.0* [114]
- *CoLAGaze* [115]
- *Reading Brain* [116]
- *RaCCooNS* [117]

**No relevant participant metadata and no relevant behavioral tasks**:

- *ADEGBTS* [118]
- *Chinese Reading* [119]
- *GazeBase* [120]
- *MECO* [121]

- *EMTeC* [122]

**Not publicly available on the web with a direct download option**:

- *Potsdam Sentence Corpus* [123]

**Low sampling rate** (below the 500 Hz threshold):

- *ETDD70* [124]
- *OASST-ETC* [125]

**Includes images** (image stimulus in addition to the textual stimulus):

- *FakeNewsPerception* [126]

## D.2 Prediction tasks

### Reader

- Language proficiency assessment [29]. Introduced with CELER, which does not meet the current dataset inclusion criteria. No other dataset with a standardized language proficiency test is currently available.
- Native language identification [26, 28]. Introduced with CELER, which does not meet the current dataset inclusion criteria. The task can be added to EyeBench using MECO L2.
- Native versus non native readers [39, 27]. Introduced with SB-SAT [39]. The task can be added to EyeBench with MECO and MECO L2.

### Reader-Text Interactions

- Ordinary reading versus information seeking [34]. Introduced with OneStop, and can be added to EyeBench.
- Ordinary reading versus text annotation [33]. Introduced with ZuCo, which does not meet the current dataset inclusion criteria. No other dataset is available for this task.
- Reading goal decoding [35]. Introduced with OneStop; can be added to EyeBench.
- First versus repeated reading [36]. Introduced with OneStop; can be added to EyeBench.
- Perceived relevance of texts [127, 128]. No public dataset is available for this task.
- Matching the reader's academic background in relation to the domain of the text [70]. Introduced with PoTeC, and can be added to EyeBench.

Note that the list above is non-exhaustive. We welcome the contributions of new tasks, as described in Appendix Section A.

