# OpenReview forum: "EyeBench: Predictive Modeling from Eye Movements in Reading"
_NeurIPS.cc/2025/Datasets_and_Benchmarks_Track — NeurIPS 2025 Datasets and Benchmarks Track poster_

### Official Review · Reviewer_8dNg · 2025-07-02

**Rating:** 4
**Confidence:** 3

**Summary:**

This paper introduces EyeBench, the first comprehensive benchmark for multimodal machine learning on eye movement data during reading. EyeBench is designed to advance research in decoding cognitive and linguistic information from gaze data paired with text. The benchmark covers a diverse range of tasks and datasets, with standardized evaluation protocols and strong baselines.

**Dataset Code Accessibility:**

Yes

**Dataset Code Comments:**

The data and code are attached in the supplementary materials.

**Ethical Considerations:**

No, there are no or only very minor ethics concerns

**Final Justification:**

I will remain the same score, 4.

**Limitations Weaknesses:**

While EyeBench draws from seven different datasets, the paper does not sufficiently detail the preprocessing procedures. Specifically:
- Are all datasets preprocessed using an identical pipeline, or are there dataset-specific adaptations?
- How are differences in raw data formats, event definitions, and coordinate systems resolved?
- Is there a unified alignment for gaze coordinates across datasets, especially considering different text layouts, screen resolutions, or experimental setups?
- Without the explanation on the above points, the conclusion drawn in the paper that 'no single model consistently outperforms others across all tasks and data splits' is less convincing.

**Strengths Contributions:**

- The paper is well-written and easy to understand, and accessible to both machine learning and cognitive science audiences.
- The benchmark provides standardized cross-validation splits, multiple generalization regimes (unseen readers, unseen texts, and both), and diverse evaluation metrics (e.g., AUROC, accuracy), allowing fair and reproducible comparisons.
- EyeBench implements and evaluates a wide range of state-of-the-art deep learning and classical machine learning models, establishing meaningful baselines for future work.

---

> ### Author Rebuttal · Authors · 2025-07-31
>
> Thank you for the insightful comments. We will address them below:
>
> EyeBench includes a diverse set of eye‑tracking corpora and introduces a single pipeline that loads, or, if missing, computes, text features, fixation‑to-aoi mappings, and tokenization for all the datasets. The preprocessing pipeline yields a unified representation for all the datasets. We preserve each dataset’s native coordinate system, event segmentation, and any provided layout metadata. To encourage cross-dataset (pre-)training rather than development on individual datasets, our approach enables training across multiple datasets. To facilitate this, EyeBench provides customizable model implementations (src/models/MODEL_NAME.py), allowing features like cross-dataset normalization to be integrated with minimal code. Following your comments, we will add a more detailed description of the preprocessing pipeline to the manuscript.

---

### Official Review · Reviewer_GSfZ · 2025-07-03

**Rating:** 5
**Confidence:** 4

**Summary:**

The EyeBench integrates and standardizes eye movement data during reading for evaluating machine learning models across multiple tasks. It incorporates datasets from multiple languages (e.g., English, German, Danish) and different task domains (e.g., cognitive assessment, lexical knowledge), which supports evaluation of model generalization over diverse populations and text types. The benchmark also encompasses various tasks in cognitive science and practical applications, enabling evaluation of models' performance across different cognitive and linguistic phenomena. It includes standardized evaluation metrics, analysis protocols, open-source tools, and provides detailed performance results of popular methods.

**Dataset Code Accessibility:**

Yes

**Ethical Considerations:**

No, there are no or only very minor ethics concerns

**Limitations Weaknesses:**

1. I would like to know if there are any experimental results of multimodal models with larger parameters based on nerual networks to make a stronger baseline. This may make this benchmark more "attractive"

**Strengths Contributions:**

1. This is 'first-of-its-kind' benchmark. It provides a good platform for comparing comprehensive tasks on a wide range of datasets.
 2. The codes and tools are accurate and easy to use, which provides more convenience for researchers in this field.

---

> ### Author Rebuttal · Authors · 2025-07-31
>
> Thank you for your careful reading of our manuscript and for your comment.
>
> Regarding your question, the 12 baseline models that we include cover state-of-the-art models for eye tracking-while-reading, with up to 355M parameters for the Transformer-based RoBERTEye model. A major challenge in training deep neural networks on eye-tracking data is the relatively limited size of available datasets, which has likely discouraged researchers from developing large-scale multimodal models until now.
> With this benchmark, we aim to encourage the community to develop new advanced models that explore different input representations, model architectures and parameter counts.

---

### Official Review · Reviewer_DCMP · 2025-07-03

**Ethics Flags:** Data privacy, copyright, and consent
**Rating:** 5
**Confidence:** 4

**Summary:**

The submission proposes to aggregate a range of datasets measuring eye movements during reading into a single, consistent benchmark for this domain. It additionally evaluates a large range of models on the benchmark, measuring the current state of the art on it.

**Dataset Code Accessibility:**

No

**Dataset Code Comments:**

The rubric for this track is different from the main NeurIPS rubric (https://neurips.cc/Conferences/2025/CallForDatasetsBenchmarks and links therein including especially https://neurips.cc/Conferences/2025/DataHostingGuidelines). In particular, it permits single-blind reviews and by doing so allows authors to put the data and code in a public place where the quality of the code/data release itself can be evaluated, ensuring that the contribution is actually usable by the community before it is accepted. The present contribution fails to do so, only including a GitHub repository in the supplement.

**I am rating this submission as a clear rejection primarily on these grounds but I think this is eminently fixable and look forward to significantly increasing my score once the authors update their submission to reflect the requirements of this track.**

EDITED: Thanks for the clarification -- agreed that if all datasets are public, just including a zip with the submission satisfies the requirements as noted in the FAQ. I'll raise my score.

**Ethical Comments:**

I appreciate the discussion of anonymization, IRBs, intended data usage, and thoughtful discouragement of identification of extra-linguistic characteristics. But then, why is the benchmark including age, gender, and other demographic information of participants in some of the datasets? I realize this is a standard in results reporting for human subjects research, included in the source datasets, and often relevant there for interpreting data in terms of generalizable conclusions and representativeness of samples. However, for the purposes of predictive benchmarking and modeling of the linguistically-relevant targets the submission identifies, I think it is not needed. Considering that reporting such characteristics is common, I don't think this is a critical concern, but scrubbing them out seems easy and would help mitigate the concern the submission raises re "prediction of extra-linguistic reader characteristics such as gender or age". I recommend that the authors revisit the data release with an eye to "what data is actually needed to evaluate performance on the benchmark".

EDITED: I agree that this is addressed thoughtfully in the submission and I agree with the ethics reviewers that including such information explicitly can help mitigate ethics concerns too (i.e. if we don't know demographic info we cannot measure when models are biased).

**Ethical Considerations:**

Yes, there are ethics concerns that require attention by the authors

**Final Justification:**

The primary issue I was concerned with was data / code availability, and I agree with the authors' clarified read of the policy, hence raising my score.

**Limitations Weaknesses:**

I have mostly minor comments here on how the work can be improved, but note that there are critical issues with the data and code release themselves I detail later in the review.

Questions
* Why binarize likerts for the STD task? It seems like the raw likerts can be used.
* Similarly for DE, why binarize the questions?


Suggestions for improvements

* From a cursory google, the EyeBench name is already used for a retinal image enhancement benchmark. Another name may be better.
* In the appendix results tables, bolding the best-performing models may help readability (and is a common convention). Or perhaps all models within the error bars of the top model could be bolded.
* Likewise for the model performance tables, it may be helpful to remind the reader whether high or low scores are best (often a latex up or down arrow is used for this).
* It may be very useful if it were possible to support arbitrary tokenizers and match extracted eye movement metrics (fixation durations etc) to those tokens (even if not words). Considering modern LLMs operate in sub-word token spaces, this may enable interesting work combining language and eye movements at the sub-word level.
* For understanding model quality at a glance, some sort of aggregate score on the benchmark would be useful in addition to per-task sub scores.
* I wonder if there's a way to report results that's not repeating RMSE in every table in sections D1 and D2 (possibly by having regression metrics in one table, and classification in the other? A wide table could be accommodated by using a landscape layout page).
* It may be useful to say a little more about the benchmark results, considering that there are many models and metrics involved. For example, do classical models ever perform well? Are there any models that do well across the board? Can the submission establish that the benchmark *is* sensitive enough to distinguish between different methods? Right now this discussion is relatively vague.
* Vague nit: some of the paper reads a little bit as "psycholinguists writing for ML people", e.g. talking about providing "an entry-point for ML researchers" into eye movement data, or "providing the ML community with a challenge". It's hard to put my finger on more specific examples but it may be worth an editing pass with the point of view of framing the paper from the perspective of the ML community -- the community is a big tent and the paper should situate itself firmly within it and not outside.

**Strengths Contributions:**

There is clear value in a "one-stop shop" for modeling eye movements in reading, and the submission does a lot to provide it by combining a number of existing datasets and tasks into a single omni-benchmark. As a bonus, it provides a reference detailing a large number of additional datasets which it chooses not to include. Furthermore, it wraps many existing models in a way that enables all-to-all evaluation across models and datasets. This sort of exhaustive and ecumenical work is great fit for the datasets and benchmarks track, and I'm optimistic that something like this can both catalyze work in the narrower field of reading research and may have a chance to attract broader ML interest. The joint motivation from research and application perspectives may help with this too, and the work provides good implementation detail. I also think that separate evaluation split on unseen reader / text / both is very good in recognizing the repeated measures nature of the data and varying difficulty of the resultant generalization tasks. I still think it may be a challenge to make this domain a standard domain for ML research, but this submission makes just about as compelling a case as I can imagine -- good job.

---

> ### Author Rebuttal · Authors · 2025-07-31
>
> Thank you very much for the detailed and constructive feedback!
>
> Please see our reply below.
>
> 1. >Datasets availability
>
> Thank you for raising this point. We believe that there has been a misunderstanding, and we would like to take this opportunity to clarify.
>
> All datasets included in the benchmark are publicly available. They are hosted by the original dataset owners on long-term storage platforms such as the Open Science Foundation (OSF). Our contribution in this aspect is to provide a data loader that automatically downloads, preprocesses, and aligns them into a standardized format.
>
> In other words, our GitHub code provides easy and direct access to the data.
> This approach fulfills the NeurIPS DB requirements regarding public data accessibility (please see the final bullet point of the “NeurIPS 2025 Datasets and Benchmarks FAQ”).
>
> We hope that our reply resolves your concern regarding data accessibility and look forward to the score update. If any additional information is needed we are happy to provide it.
>
> 2. >Ethical considerations
>
> Thank you for raising this point. The extra-linguistic participant characteristics are publicly available and hosted online as part of the data releases by the original dataset authors. However, we agree with your concern and will exclude these fields when the data is accessed via the benchmark scripts.
>
> 3. >Binarization of STD and DE
>
> Thank you, this is an excellent point. Following the paper that introduced the STD task we used a binarized version of the task. However, we agree that modeling this task on the full Likert scale is more appropriate. We will therefore introduce a new variant of the task with prediction of Likert scores and will report the results in the paper. For completeness, and to enable comparison with prior work, we will also report binary results by thresholding the Likert predictions.
>
> For the DE task on PoTeC, we chose to binarize the labels because the questions used to assess domain expertise (DE) are not standardized and vary in difficulty across texts, resulting in non-comparable label scales.
>
> 4. >Additional suggestions
>
> Following your suggestions we will:
>
> * Consider other names for the benchmark to avoid confusions.
> * Mark in bold the best-performing models.
> * Indicate whether high or low scores are the best.
> * Add a table with aggregated scores for the regression and classification tasks for the main text.
> * Separate regression and classification metrics into two tables in the appendix.
> * Expand on the results section.
> * Edit the writing to better frame the paper in the context of ML research and the ML community.
>
> 5. >Arbitrary tokenizers
>
> Finally, we note that our framework already supports matching text tokenized by arbitrary tokenizers (supported by HuggingFace AutoTokenizer) with fixation- and word-level eye movement information. Currently this is implemented as part of specific models. We will clarify this in the main text, and will add a technical description in the appendix.
>
> Many thanks again for your very thoughtful engagement with our work!

---

### Official Review · Reviewer_LBka · 2025-07-07

**Rating:** 4
**Confidence:** 4

**Summary:**

This paper introduces EyeBench, the first unified benchmark for multimodal modeling of eye movements and text during reading. EyeBench aggregates seven public eye‑tracking datasets covering two problem categories and standardizes data loading, preprocessing, and evaluation. EyeBench aims to accelerate progress in reading‑behavior modeling, guiding future work on better alignment, larger-scale data, and improved fusion strategies.

**Dataset Code Accessibility:**

Yes

**Ethical Considerations:**

No, there are no or only very minor ethics concerns

**Final Justification:**

Thanks for your reply. I have read the rebuttal, and most of my concerns are solved. Therefore, I keep my positive score.

**Limitations Weaknesses:**

The work is largely engineering‑oriented, with few novel design elements. It does not construct or define some new task scenarios or paradigms, relying instead on task settings that already exist. Additionally, it does not introduce any new datasets, using only previously public data.

The paper mentions that analyzing eye-tracking data during reading presents unique challenges compared to other multimodal tasks. However, aside from the introduction of the three generalization settings, EyeBench does not appear to incorporate specific features that directly address these particular challenges. Including more evaluation strategies tailored to those difficulties would make the work more thorough and substantial.

Although the backgrounds and significance of the various tasks are described in comprehensive detail, the length of this content forces many tables into the appendix, preventing them from standing out in the main text. Condensing some of the earlier prose and moving key tables into the body of the paper could address this issue.

I have some questions about the Domain Expertise (DE) task: the paper claims that this task is introduced in EyeBench for the first time. However, if that is indeed the case, how could a pre‑existing dataset, i.e., the PoTeC dataset, be used to support this task? Where did the labels come from? If these labels were provided by the original database, doesn’t that imply the database already incorporated this task when it was constructed?

**Strengths Contributions:**

The paper is clearly written, with a thorough description of its background and significance. It is the first benchmark for analyzing eye‑tracking data during reading, which will facilitate future research by providing a unified evaluation protocol and enhancing fairness.

---

> ### Author Rebuttal · Authors · 2025-07-31
>
> Thank you for your careful reading of our manuscript and the detailed comments, which we address below:
>
> 1. >The work is largely engineering‑oriented, with few novel design elements.
>
> Thank you for this comment – It helped us recognize the need to more clearly emphasize the novel tasks and task formulations introduced in our work. For instance, we introduce tasks that have not previously been studied such as Domain Expertise by utilizing PoTeC and the prediction of reading comprehension on the CopCo dataset.
>
> Furthermore, we propose extensions of existing tasks that better capture the fine-grained nature of the underlying cognitive processes. In the case of subjective text difficulty (STD) on SB-SAT, as also mentioned in our response to Reviewer DCMP, we go beyond the conventional binary classification paradigm, and introduce a multi-class formulation that predicts Likert-scale scores. Compared to prior work, this approach provides a more granular understanding of how eye movement patterns relate to perceived difficulty.
>
> We appreciate the opportunity to clarify these contributions and will revise the manuscript accordingly to highlight them more explicitly.
>
> 2. >Unique challenges compared to other multimodal tasks.
>
> Thank you for raising this point. EyeBench includes additional components tailored to the challenges of eye tracking for reading data and related applications.
> First, we provide code for data preprocessing that handles the specific structure and formatting of eye-tracking data collected during reading, facilitating experimentation and replication.
>
> Second, the models included in EyeBench were specifically developed to represent eye movements and integrate them with text.
> Each model addresses the challenges inherent to the multimodal nature of eye-tracking-while-reading data in different ways. For example, a key distinction lies in how the models align the static text (stimulus) with the chronological sequence of gaze data. We additionally introduce text only and reading speed baselines which are essential for demonstrating the utility of eye movement information for the addressed tasks.
>
> Finally, the mentioned new subject/item/subject+item evaluations address not only the structured non-i.i.d. nature of the data, but also reflect use cases of real-world applications where training data availability varies—such that prior eye-tracking data for the target participants and texts may or may not be available.
>
> Overall, we would like to emphasize that identifying optimal representations and modeling strategies for eye movements remains an open research question. EyeBench contributes to this effort by offering a standardized framework for systematically evaluating and comparing different approaches. We will revise the manuscript to make these contributions and their relevance to the stated challenges more explicit.
>
> 3. >Condensing and moving key tables to main
>
> Thank you for this suggestion. We will condense the task descriptions and add a summary table with average results per model to the main text.
>
> 4. >Domain Expertise (DE) task
>
> DE is indeed introduced in EyeBench for the first time as a prediction task. The PoTeC dataset is the result of a major data collection effort, with the main output being the data itself. While the relevant labels are part of the data release, the task itself was not introduced nor addressed by the PoTeC authors (see Jakobi et al., 2024, Behavior Research Methods).

---

### Decision · Program_Chairs · 2025-09-18

**Decision:**

Accept (poster)

**Comment:**

This paper introduces EyeBench, a novel benchmark for evaluating multimodal machine learning models that analyze eye movements during reading. The benchmark aims to foster innovation in AI and cognitive science by providing a standardized framework for diverse datasets and tasks, such as predicting reading comprehension and assessing text difficulty.

The paper received four reviews, with final ratings of 4 (Borderline accept), 5 (Accept), 5 (Accept), and 4 (Borderline accept).

The main strengths of the paper include its novelty as the first unified benchmark for eye-tracking data in reading, its clear writing, and its comprehensive aggregation of existing datasets and models. The structured evaluation settings for unseen readers and texts are also a notable strength. A key weakness initially raised was the lack of novel design elements or new datasets, and insufficient detail on preprocessing procedures. Another significant concern was the initial non-compliance with data and code accessibility requirements. The authors addressed these concerns by clarifying the novelty of their task formulations, promising more detailed preprocessing descriptions, and confirming that all datasets are publicly available with their code providing automated access, thereby satisfying the accessibility requirements.

Given the strong positive reception from reviewers after the rebuttal period, and the authors' effective addressing of all major concerns, the paper is recommended for acceptance.